# Orexin neurons play contrasting roles in itch and pain neural processing via projecting to the periaqueductal gray
Tatsuroh Kaneko [1] ✉, Asuka Oura[1], Yoshiki Imai[1], Ikue Kusumoto-Yoshida[1], Takuro Kanekura[2], Hiroyuki Okuno [3], Tomoyuki Kuwaki [1] & Hideki Kashiwadani [1] ✉

Pain and itch are recognized as antagonistically regulated sensations; pain suppresses itch, whilst pain inhibition enhances itch. The neural mechanisms at the central nervous system (CNS) underlying these pain-itch interactions still need to be explored. Here, we revealed the contrasting role of orexin-producing neurons (ORX neurons) in the lateral hypothalamus (LH), which suppresses pain while enhancing itch neural processing, by applying optogenetics to the acute pruritus and pain model. We also revealed that the circuit of ORX neurons from LH to periaqueductal gray regions served in the contrasting modulation of itch and pain processing using optogenetic terminal inhibition techniques. Additionally, by using an atopic dermatitis model, we confirmed the involvement of ORX neurons in regulating chronic itch processing, which could lead to a novel therapeutic target for persistent pruritus in clinical settings. Our findings provide new insight into the mechanism of antagonistic regulation between pain and itch in the CNS.

Pain and itch are aversive sensations, eliciting distinct defense responses to ward off external threats. Specifically, pain triggers a withdrawal reflex to avert additional harm, while itch incites a scratching response to rid the skin surface of irritants. These two sensations have traditionally been viewed as antagonistically controlled phenomena, with pain inhibiting itch and, conversely, itch being provoked by the suppression of pain[1,2].

The underlying neural mechanisms of these pain-itch interactions have been explored, with significant findings revealing their operation at the spinal cord level[3–8]. However, despite the progress made at the spinal cord level, minimal research exists thus far regarding the involvement of supraspinal regions in the neural processing of these interactions[2].

Recent research has underscored the crucial role of the ventrolateral periaqueductal gray (vlPAG) region in modulating pain and itch processing, with vlPAG-glutamatergic and -GABAergic neurons playing contrasting roles in these processes[9,10].

At a higher CNS level than the PAG, we made the novel discovery that orexin-producing neurons (ORX neurons) exhibit opposing roles in pain and itch neural processing, with inhibition of pain but facilitation of itch neural processing[11]. However, our prior study employing ORX neuron-ablated mice (ORX-abl mice) did not decipher the precise neural circuit of ORX neurons in itch and pain processing.

In this study, we used optogenetics to manipulate the activity of ORX neurons in real-time to assess the function and the precise neural circuit of ORX neurons in itch and pain processing and bolstered our previous findings. We identified that the neural circuit of ORX neurons from the lateral hypothalamus (LH) to the PAG area contributes to the bidirectional modulation of itch and pain processing using optogenetic terminal inhibition techniques. In addition, we evaluated the clinical implications of our findings by confirming the involvement of ORX neurons in chronic itch processing using an atopic dermatitis-related model.

Our results offer a novel viewpoint for understanding the antagonistic regulation of itch and pain in the central nervous system.

## Results

### The optical inhibition of orexin neurons elicited a contrasting modulation in itch and pain

Firstly, we employed an optogenetic strategy to modulate the activity of ORX neurons in an acute itch model (chloroquine-induced at the nape) and a pain model (capsaicin-induced at the cheek). We selectively induced the expression of Archaerhodopsin-T (ArchT) in ORX neurons in the lateral hypothalamus (LH) through an injection of AAV-TRE-ArchT-mCherry into the LH area of ORX-tTA mice, followed by the implantation of an

[1]Department of Physiology, Graduate School of Medical and Dental Sciences, Kagoshima University, Kagoshima, Japan. [2]Department of Dermatology, Graduate School of Medical and Dental Sciences, Kagoshima University, Kagoshima, Japan. [3]Laboratory of Biochemistry and Molecular Biology, Graduate School of Medical and Dental Sciences, Kagoshima University, Kagoshima, Japan. ✉e-mail: k9357850@kadai.jp; danny@m3.kufm.kagoshima-u.ac.jp

optical fiber in the LH after 3 weeks (Fig. 1a). As a control, AAV-TRE-mCherry was utilized. Upon completion of the behavioral experiments, all subjects were euthanized, and we confirmed that most of the ArchT-mCherry expressions were localized within ORX neurons (91.8 ± 0.5%; Fig. 1b), and also fiber implantation sites were confirmed in all animals (Supplementary Fig. 1a). We also verified that applying green light (wavelength 532 nm; continuous; 8–10 mW; 3-min intervals; for 30 min) did not modify scratching or wiping behavior without itch or pain stimuli (Fig. 1c).

In a time-course analysis of scratching bouts, we found that optogenetic inhibition of ORX neurons through the implanted optical fibers in freely moving animals resulted in a reduction in scratching behaviors in the chloroquine-induced pruritus model (Fig. 1d, e).

A two-way ANOVA coupled with Tukey's multiple comparisons test demonstrated a significant decrease in the number of scratching events during the observation period (30 min) with the optogenetic inhibition of ORX neurons ($Scratch_{mCherry/Light-OFF} = 62.8 \pm 5.6$, $Scratch_{mCherry/Light-ON} = 66.5 \pm 6.6$, $Scratch_{ArchT/Light-OFF} = 84.0 \pm 9.2$, $Scratch_{ArchT/Light-ON} = 34.1 \pm 4.5$, $p_{mCherry/Light-ON \text{ vs. } ArchT/Light-ON} = 0.0149$, $p_{ArchT/Light-OFF \text{ vs. } ArchT/Light-ON} = 0.0001$; Fig. 1f).

Conversely, in the pain experiment, the optogenetic inhibition of ORX neurons led to a significant escalation in pain-associated behavior (wiping) in the capsaicin-induced pain model ($Wiping_{mCherry/Light-OFF} = 30.5 \pm 11.5$, $Wiping_{mCherry/Light-ON} = 35.3 \pm 11.4$, $Wiping_{ArchT/Light-OFF} = 27.3 \pm 10.8$, $Wiping_{ArchT/Light-ON} = 100.6 \pm 19.5$, $p_{mCherry/Light-ON \text{ vs. } ArchT/Light-ON} = 0.0119$, $p_{ArchT/Light-OFF \text{ vs. } ArchT/Light-ON} = 0.0045$; Fig. 1g–i).

These findings align with our previous study employing ORX-abl mice[11]. Consequently, these results from optogenetic experiments bolster the evidence for the diametric modulation by ORX neurons in itch and pain processing: enhancing itch while suppressing pain.

### The optical activation of orexin neurons unaltered itch and pain response

Subsequently, we evaluated the impact of optically activating ORX neurons on itch and pain processing. This was done by injecting AAV-TRE-channelrhodopsin-2(ChR2)-mCherry into the LH area of ORX-tTA mice, followed by the implantation of an optical fiber in the LH after 3 weeks (Fig. 2a). AAV-TRE-mCherry served as the control. After the behavioral experiments, all subjects were euthanized, and the expression of the virus in ORX neurons was histologically verified (91.9 ± 0.8%; Fig. 2b), and the fiber implantation sites were confirmed (Supplementary Fig. 1b). We also ascertained that applying blue light pulses alone (10 ms; 20 Hz; 6–8 mW; 3-min intervals; for 30 min) did not affect the scratching and wiping behaviors of naïve mice (Fig. 2c).

The optogenetic activation of ORX neurons did not influence the scratching behaviors in the chloroquine-induced pruritus model ($Scratch_{mCherry/Light-OFF} = 71.4 \pm 13.4$, $Scratch_{mCherry/Light-ON} = 74.6 \pm 9.6$, $Scratch_{ChR2/Light-OFF} = 74.8 \pm 13.0$, $Scratch_{ChR2/Light-ON} = 60.0 \pm 10.0$, $p_{mCherry/Light-ON \text{ vs. } ChR2/Light-ON} = 0.8239$, $p_{ChR2/Light-OFF \text{ vs. } ChR2/Light-ON} = 0.804$; Two-way ANOVA with Tukey's multiple comparisons tests; Fig. 2d, e). Moreover, the optogenetic activation of ORX neurons also did not affect the pain-associated behavior in the capsaicin-induced pain model ($Wiping_{mCherry/Light-OFF} = 18.9 \pm 6.8$, $Wiping_{mCherry/Light-ON} = 32.8 \pm 11.0$, $Wiping_{ChR2/Light-OFF} = 32.0 \pm 8.5$, $Wiping_{ChR2/Light-ON} = 34.8 \pm 8.5$, $p_{mCherry/Light-ON \text{ vs. } ChR2/Light-ON} = 0.9983$, $p_{ChR2/Light-OFF \text{ vs. } ChR2/Light-ON} = 0.9954$; Fig. 2f, g).

There is a concern that ChR2-induced optical stimulation might not work precisely in our experiment. Therefore, to confirm the activation level of ORX neurons by ChR2-induced optical stimulation, we evaluated the response of ORX neurons to optical stimulation using c-Fos as a neuronal activity marker (Supplementary Fig. 2).

Immunohistochemical analyses revealed that the ratio of c-Fos-positive ORX neurons to the total ORX neurons was significantly increased after optical stimulation ($R_{Light \, OFF} = 13.4\% \pm 1.1\%$, $R_{Light \, ON} = 65.9\% \pm 2.8\%$, $p < 0.001$; Supplementary Fig. 2b, c). The activation level of ORX neurons by optical stimulation is almost equal to the result by pruritic

stimulation with chloroquine, which was confirmed in our previous study ($R_{chloroquine} = 55.1\% \pm 10.4\%$)[11].

These findings suggest the potential that the stimulation of pruritogen or algogen fully activates ORX neurons involved in itch and pain processing. Further activation of ORX neurons via optogenetics may be ineffective due to the saturation of ORX neuron activation by itch or pain stimulation.

### Inhibition of the LH → PAG axis of orexin neurons induced an opposite modulation in itch and pain

ORX neurons have a wide range of projections from the LH to various brain regions such as the ventral tegmental area (VTA), locus coeruleus (LC), dorsal raphe, PAG, and so on[12]. This raises the question of which ORX neural circuit is pivotal for their contrasting modulation in pain and itch processing.

In our previous work, we noted that the elevation of pruritogen-induced c-Fos expression in neurons located in the lateral and ventrolateral parts of the periaqueductal gray (lPAG and vlPAG, respectively) was diminished in ORX neuron-ablated mice compared to the WT control[11]. This indicated that the synaptic output of ORX neurons to the PAG area might be important in the contrasting modulation for itch and pain processing.

To directly scrutinize the significance of the LH → PAG projection of ORX neurons, we undertook an optogenetic terminal inhibition experiment[13].

Initially, we injected AAV-TRE-ArchT-mCherry into the LH of ORX-tTA mice and then positioned an optical fiber over the lPAG/vlPAG area (Fig. 3a). AAV-TRE-mCherry was utilized as a control. After the completion of the behavioral experiments, all subjects were euthanized, and it was confirmed that most of the ArchT-mCherry expressions were localized within ORX neurons in the LH (95.9 ± 0.8%; Fig. 3b left), and the axons of ArchT-mCherry expressing neurons were present in the lPAG/vlPAG area (Fig. 3b right). We also confirmed that the position of the optical fiber was located over the lPAG/vlPAG area (Supplementary Fig. 1c).

Before the itch- and pain-behavioral experiment, we verified that merely applying green light (wavelength 532 nm; continuous; 8–10 mW; 3-min intervals; for 30 min) to the lPAG/vlPAG area did not alter the scratching and wiping behaviors in naïve animals (Fig. 3c).

The mice with optogenetically inhibited $LH^{ORX} \rightarrow$ lPAG/vlPAG neuronal terminals exhibited a substantial decrease in chloroquine-induced scratching behavior ($Scratch_{mCherry/Light-OFF} = 70.0 \pm 5.8$, $Scratch_{mCherry/Light-ON} = 59.3 \pm 8.1$, $Scratch_{ArchT/Light-OFF} = 72.7 \pm 14.0$, $Scratch_{ArchT/Light-ON} = 18.6 \pm 3.1$, $p_{mCherry/Light-ON \text{ vs. } ArchT/Light-ON} = 0.0121$, $p_{ArchT/Light-OFF \text{ vs. } ArchT/Light-ON} = 0.0011$; Fig. 3d, e). Conversely, the optogenetic inhibition of ORX neurons at lPAG/vlPAG neuronal terminals significantly escalated the pain-related behavior (wiping) in the capsaicin-induced pain model ($Wiping_{mCherry/Light-OFF} = 16.5 \pm 6.4$, $Wiping_{mCherry/Light-ON} = 15.2 \pm 6.3$, $Wiping_{ArchT/Light-OFF} = 10.6 \pm 4.5$, $Wiping_{ArchT/Light-ON} = 68.0 \pm 16.7$, $p_{mCherry/Light-ON \text{ vs. } ArchT/Light-ON} = 0.0024$, $p_{ArchT/Light-OFF \text{ vs. } ArchT/Light-ON} = 0.0013$; Fig. 3f, g).

These observations are consistent with the experiment of optogenetic inhibition at the cell body of ORX neurons in the LH (Fig. 1). Thus, the results from the current optogenetic terminal inhibition experiment suggest that the projection of ORX neurons from LH to lPAG/vlPAG (LH → PAG axis) is crucial in the contrasting modulation of itch and pain processing.

### Orexin neurons are involved in the regulation of chronic itch processing

Chronic itch is linked with elevated stress, anxiety, and other mood-related disorders. These conditions aggravate itch symptoms, establishing a harmful cycle that deteriorates the prognosis of the condition and impairs the quality of life[14].

While most previous studies have concentrated on peripheral and spinal mechanisms of chronic itch[15–17], our understanding remains limited regarding the mechanism in the central nervous system[18,19].

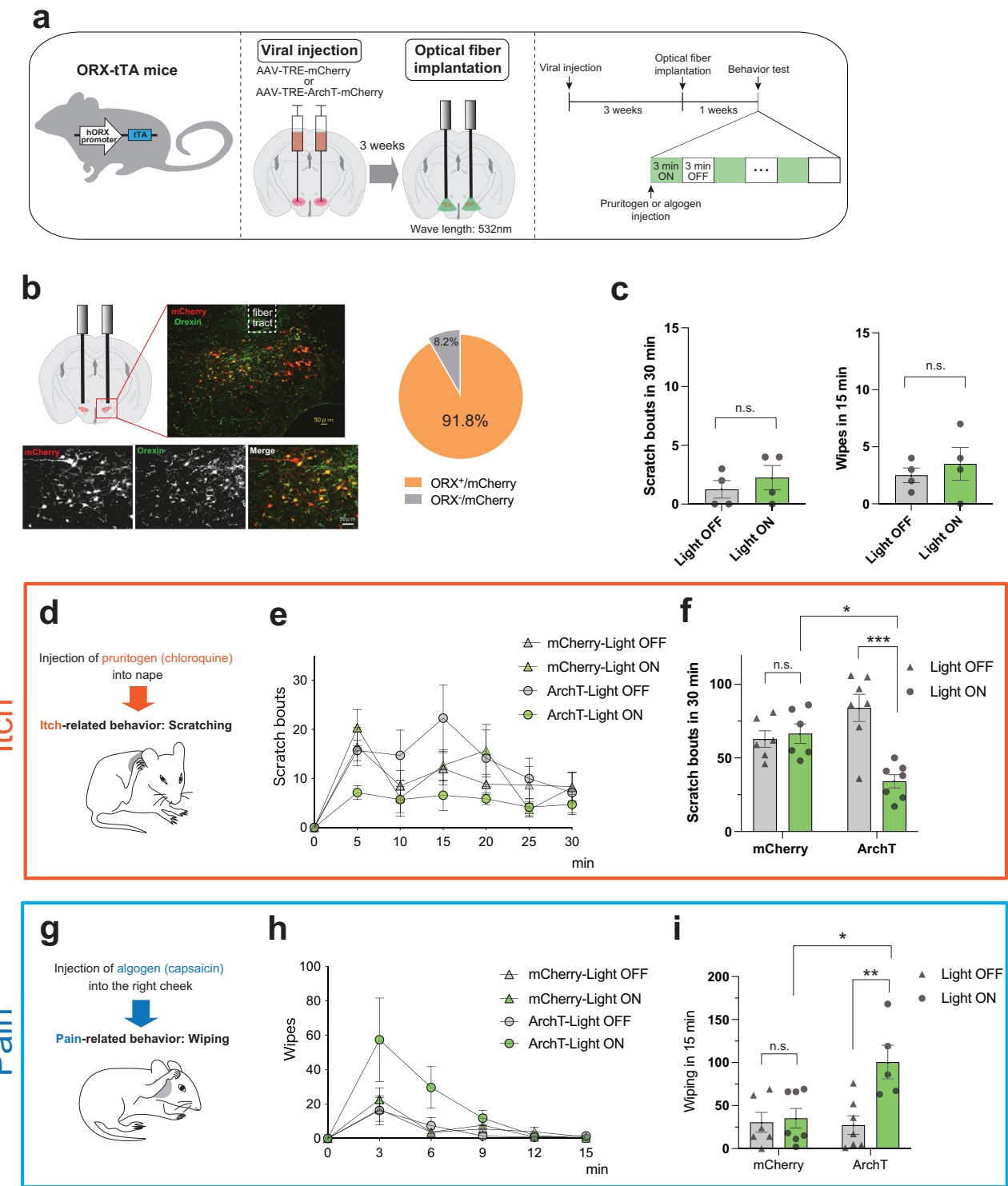

**Fig. 1 | Optical inhibition of ORX neurons induces opposite modulation in itch and pain. a** Left, schematic illustration of ORX-tTA mice. Middle, schematic showing the injection of the AAV-TRE-ArchT-mCherry into the LH of ORX-tTA mice and implantation of the optical fiber. Right, the timeline of the experiments. **b** Left, Histological verification of viral injection and optical fiber implantation in a representative ORX-tTA mouse. Right, the quantification (inset pie chart) shows the co-expression of ArchT-mCherry with orexin in LH of ORX-tTA mice ($n = 3$ sections per animal from 7 animals). Scale bars, 50 μm. **c** Optical inhibition of ORX neurons did not alter scratching behaviors in the absence of pruritogen (left) and wiping behaviors in the absence of algogen (right) ($n = 4$ mice in each group). Paired Student's $t$ test. **d** Schematic illustration of scratching (itch-related behavior) induced by the chloroquine injection into the nape. **e** Time course of the number of

scratching behaviors induced by intradermal chloroquine injection. The plots indicate the cumulative number of scratching bouts recorded every 5 min. **f** Optical inhibition of ORX neurons reduced the chloroquine-induced scratching behaviors ($n = 6–7$ mice in each group). (**g**) Schematic illustration of wiping (pain-related behavior) induced by the capsaicin injection into the right cheek. **h** Time course of the number of wiping behaviors induced by capsaicin injection into the cheek. The plots indicate the cumulative number of wiping events recorded every 3 min. **i** Optical inhibition of ORX neurons increased the capsaicin-induced wiping behaviors ($n = 5–7$ mice in each group). The data represent the mean ± SEM. $*p < 0.05$, $**p < 0.01$, $***p < 0.001$; two-way ANOVA with Tukey's multiple comparisons test. hORX promoter, human prepro-orexin promoter; tTA, tetracycline transactivator; TRE, tetracycline response element; LH, lateral hypothalamus.

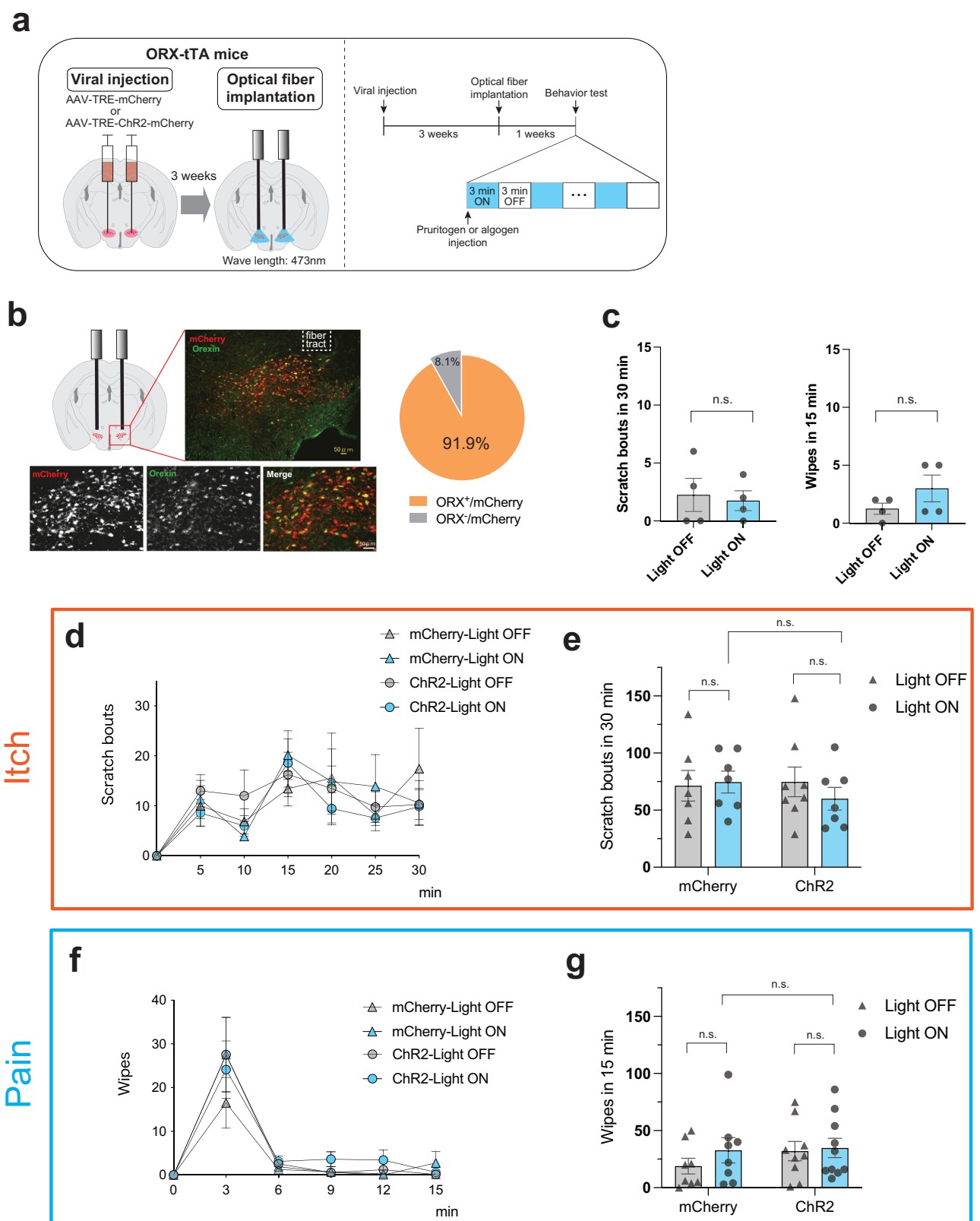

To investigate whether ORX neurons also participate in chronic itch processing, we utilized ORX neuron ablation mice (ORX-abl mice, Fig. 4a) and created a model of contact dermatitis-associated chronic itch by repeatedly applying diphenylcyclopropenone (DCP) to the nape skin of mice, as described earlier[15,18,20,21] (Fig. 4b). We confirmed that ORX neurons

were completely ablated in ORX-abl mice compared to WT mice (Supplementary Fig. 3).

In the DCP-induced chronic itch model, ORX-abl mice exhibited significantly reduced scratching behavior compared to their wild-type (WT) counterparts during 60 minutes of the observation period

**Fig. 2 | Optical stimulation of orexin neurons does not affect itch and pain. a** Left, schematic showing the injection of the AAV-TRE-ChR2-mCherry into the LH of ORX-tTA mice and implantation of the optical fiber. Right, the timeline of the experiments. **b** Left, Histological confirmation of viral injection and optical fiber implantation in a representative ORX-tTA mouse. Right, the quantification (inset pie chart) shows the co-expression of ChR2-mCherry with orexin in LH of ORX-tTA mice (*n* = 3 sections per animal from 10 animals). Scale bars, 50 μm. **c** Optical activation of ORX neurons did not alter scratching behaviors in the absence of pruritogen (left) and wiping behaviors in the absence of algogen (right) (*n* = 4 mice in each group). Paired Student's *t* test. **d** Time course of the number of scratching behaviors induced by intradermal chloroquine injection. The plots indicate the cumulative number of scratching bouts recorded every 5 min. **e** Optical activation of ORX neurons did not alter the chloroquine-induced scratching behaviors (*n* = 7–8 mice in each group). **f** Time course of the number of wiping behaviors induced by capsaicin injection into the cheek. The plots indicate the cumulative number of wiping events recorded every 3 min. **g** Optical activation of ORX neurons did not alter the capsaicin-induced wiping behaviors (*n* = 8–10 mice in each group). The data represent the mean ± SEM. Two-way ANOVA with Tukey's multiple comparisons test was done for statistical analysis. ORX, orexin; tTA, tetracycline transactivator; TRE, tetracycline response element; LH, lateral hypothalamus.

(Scratch$_{WT/DW}$ = 21.3 ± 3.5, Scratch$_{WT/DCP}$ = 320.4 ± 43.0, Scratch$_{ORX-abl/DW}$ = 6.4 ± 5.4, Scratch$_{ORX-abl/DCP}$ = 70.7 ± 15.7, $p_{WT/DW \text{ vs. } WT/DCP}$ < 0.0001, $p_{WT/DCP \text{ vs. } ORX-abl/DCP}$ < 0.0001; Fig. 4c, d). These findings imply that ORX neurons also have a crucial role in chronic itch processing, not just in acute pruritus.

We also measured transepidermal water loss (TEWL), a well-established metric for assessing skin barrier function[22]. Notably, ORX-abl mice demonstrated significantly lower TEWL compared to the WT control group (TEWL$_{WT/DW}$ = 2.5 ± 0.3, TEWL$_{WT/DCP}$ = 29.2 ± 3.0, TEWL$_{ORX-abl/DW}$ = 0.6 ± 0.2, TEWL$_{ORX-abl/DCP}$ = 16.8 ± 1.2, $p_{WT/DW \text{ vs. } WT/DCP}$ < 0.0001, $p_{WT/DCP \text{ vs. } ORX-abl/DCP}$ = 0.0076; Fig. 4e).

This suggests that inhibiting ORX neurons may mitigate scratching behavior and consequently prevent skin barrier dysfunction induced by chronic itch. Thus, ORX neurons contribute to itch neural processing similarly under both acute and chronic itch conditions.

## Discussion

The interplay between sensations of pain and itch is widely recognized as antithetical; pain acts as a suppressor for the itch while inhibiting pain can induce itch. Despite this, evidence remains scarce regarding how this pain-itch interplay is neurologically processed within the central nervous system. In this study, we applied an optogenetic approach to shed light on the diametric roles of ORX neurons in processing itch and pain (Figs. 1, 2). Furthermore, using optogenetic terminal inhibition techniques, we discovered that the neural circuit of ORX neurons from LH to the PAG region plays a crucial part in this bidirectional modulation of itch and pain processing (Figs. 3, 5). We also examined the involvement of ORX neurons in the processing of chronic itch, utilizing a chronic contact dermatitis model, which suggests its potential as a novel therapeutic target for chronic pruritus (Fig. 4).

Orexin neuropeptides A and B, also known as hypocretin-1 and -2, originate from the precursor peptide pre-pro-orexin[23]. The brain hosts two specific orexin receptors—orexin type-1 receptor (ORX-1R) and orexin type-2 receptor (ORX-2R)—which manifest unique expression profiles across different areas[24]. The ORX neural circuit plays an instrumental role in an array of physiological responses such as arousal, reward-oriented behavior, energy balance, sensory modulation, stress management, cognitive function, and endocrine control[25]. While ORX neurons serve multiple functions, our laboratory has predominantly centered on their role in managing the body's defense response to external stressors[26,27]. This defensive mechanism gears the body for a fight-or-flight response in the face of external stressors, triggering autonomic changes, including an increase in blood pressure, heart rate, and respiration, and producing an analgesic effect against pain stress. Previous findings from our laboratory have delineated the role of ORX neurons in instigating the various efferent pathways involved in this defense response. As one of the defense responses, our prior work[11] has indicated the possibility that ORX neurons exhibit the opposite roles in pain and itch neural processing, whereby they suppress pain yet facilitate itch. However, to examine the function of ORX neurons in pain and itch processing, we used ORX neuron ablated mice (ORX-abl mice), which did not allow us to address the precise neural circuit of ORX neurons in this processing. In addition, the ablation of ORX neurons started from birth, and almost ORX neurons were ablated at 4 weeks of age[28]. This poses the potential concern that the ablation of ORX neurons might instigate alterations in the development of other neural networks, possibly influencing our experimental outcomes. To overcome these limitations, we adopted optogenetic techniques to investigate further the contrasting modulation of itch and pain by ORX neurons[29]. Our results revealed that optically inhibiting ORX neurons reduced itch while amplifying pain (Fig. 1). This observation aligns with our previous research using ORX-abl mice[11], thereby reinforcing the assertion that ORX neurons exert opposing influences on itch and pain processing—promoting itch while inhibiting pain, both of which could be interpreted as defense responses triggered by ORX neurons.

Intriguingly, in our optogenetic activation experiment utilizing channelrhodopsin-2 (ChR2), the optogenetic activation of ORX neurons did not affect scratching behaviors in the chloroquine-induced pruritus model nor pain-associated behavior in the capsaicin-induced pain model (Fig. 2). There is a possibility that ChR2-induced optical stimulation might not be worked precisely. However, we confirmed the specific expression of ChR2-mCherry in ORX neurons and the correct positions of optical fibers (Fig. 2b, Supplementary Fig. 1b), and the parameters of optical stimulation in the present experiment had worked in our previous report[30]. Neuronal functionality is dictated by certain physiological boundaries. For instance, there is an upper threshold to the frequency of action potential generation, which is influenced by factors such as the refractory period (the time following an action potential during which a neuron cannot fire again)[31]. Provided that a specific neuronal population is already firing at its maximum rate in response to existing non-optogenetic stimulation (such as the presence of a chemical stimulant in the present study), further optogenetic stimulation might not enhance firing rates anymore. In line with this hypothesis, we confirmed that the activation level of ORX neurons by ChR2-induced optical stimulation was almost equal to the level by pruritic stimulation in our previous report[11] (Supplementary Fig. 2). Thus, in the context of ORX neurons involved in itch and pain processing, if these neurons are already maximally activated by pruritogens or algogens, additional optogenetic activation might be ineffective due to saturation of neuronal activation. Such a potential ceiling effect may explain why optogenetic activation of ORX neurons failed to modulate both itch and pain processing in our experiments.

The neurons located in the PAG region have been reported to play contrasting roles in pain and itch neural processing[9,10]; In pain processing, nociception was suppressed upon activation of PAG glutamatergic neurons, with its potentiation occurring upon their inhibition[9]. In itch processing, activation and inhibition of these neurons correspondingly lead to enhanced itch-related behaviors and suppressed itch, respectively[10]. Our results concerning ORX neurons are allied with these observations. Past anatomical studies have revealed the projections of ORX neurons to the PAG[32,33] and identified both ORX-1R and -2R expression within the PAG region[24]. Moreover, the activation of ORX neurons in the LH during stress conditions initiates orexin release, which induces an analgesic effect via ORX-1R in the vlPAG[34]. Our preceding research[11] demonstrated a reduction in pruritogen-induced c-Fos expression in PAG neurons in ORX-abl mice as compared to wild-type controls, suggesting that the LH → PAG projection of ORX neurons potentially plays a part in facilitating itch neural processing, in addition to its role in stress-induced analgesia in pain processing. In the present study, to more closely examine the potential significance of the LH → PAG projection of ORX neurons, we utilized an optogenetic terminal inhibition experiment[13]. In this method, by combining opsins localized

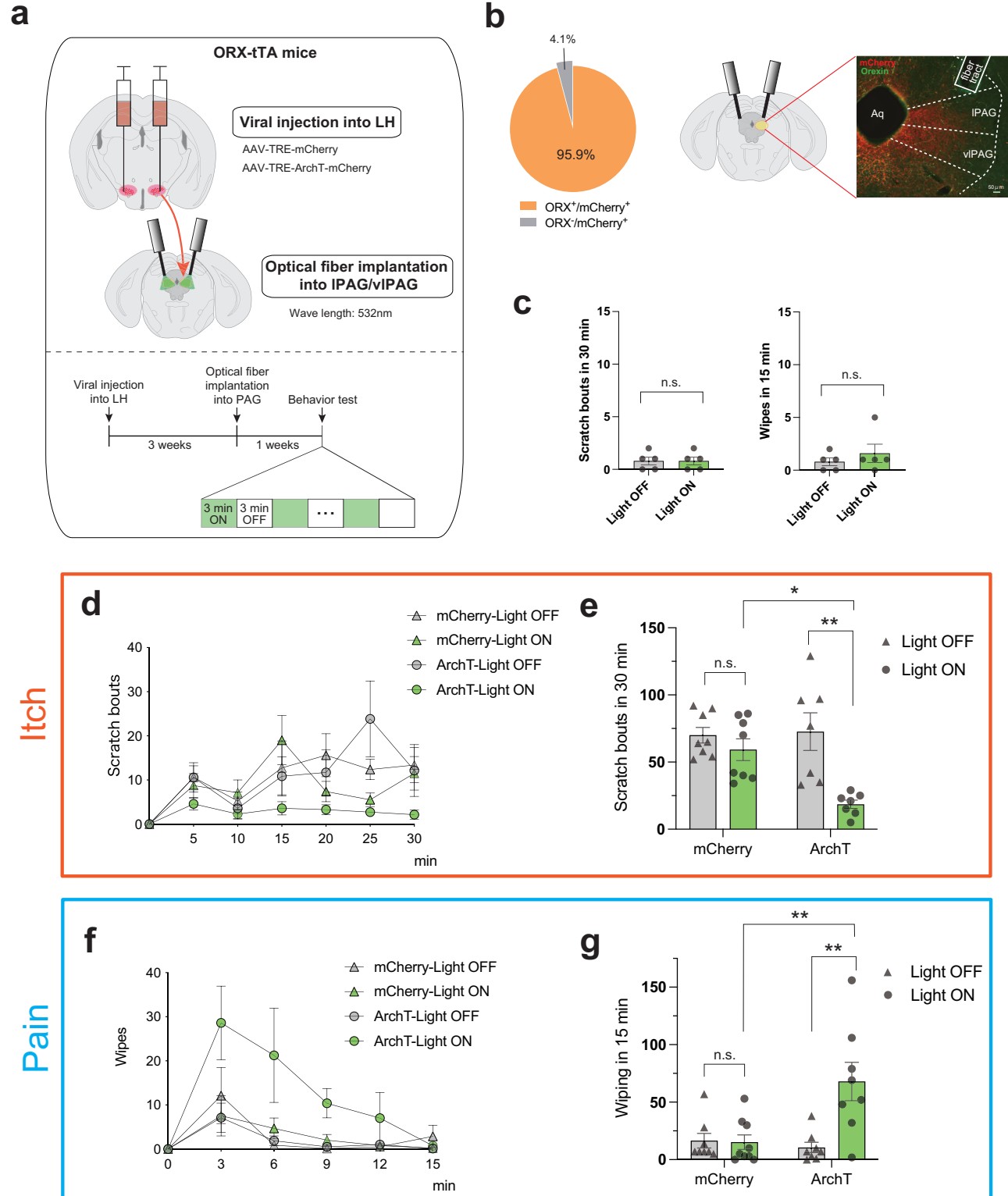

presynaptically with light stimulation at the terminals, we were able to investigate the intricate neural wiring of the brain by probing functional connectivity directly at the level of specific synapses; light-driven excitation or inhibition of axon terminals could reveal synaptic connections between two neuronal populations. Through this method, we revealed that ArchT-mediated optical inhibition of $LH^{ORX} \rightarrow lPAG/vlPAG$ neuronal terminals reduced itch-related behavior and enhanced pain-related behavior (Fig. 3). These results align with those obtained from optogenetic inhibition at the

cell body of ORX neurons in the LH (Fig. 1). Consequently, the optogenetic terminal inhibition experiment points to a pivotal role for the $LH \rightarrow PAG$ axis projection of ORX neurons in the opposing modulation of itch and pain processing (Fig. 5).

Gao and colleagues discovered that subpopulations of PAG glutamatergic neurons might segregate pain and itch neural processing, including tachykinin 1 (Tac1)-expressing and somatostatin (SST)-expressing neurons[35]. Their research confirmed that the elimination or pharmacogenetic

**Fig. 3 | Optical inhibition of the LH → PAG axis of orexin neurons induces opposite modulation in itch and pain. a** Top, schematic showing the injection of the AAV-TRE-ArchT-mCherry into the LH of ORX-tTA mice and implantation of the optical fiber into lPAG/vlPAG. Bottom, the timeline of the experiments. **b** Left, the quantification (inset pie chart) shows the co-expression of ArchT-mCherry with orexin in LH of ORX-tTA mice (*n* = 3 sections per animal from 8 animals). Right, histological confirmation of ArchT-mCherry expression and optical fiber implantation at lPAG/vlPAG in a representative ORX-tTA mouse. Scale bars, 50 μm. **c** Optical inhibition of LH^ORX → lPAG/vlPAG neuronal terminals did not alter scratching behaviors in the absence of pruritogen (left) and wiping behaviors in the absence of algogen (right) (*n* = 5 mice in each group). Paired Student's *t* test. (**d**) Time course of the number of scratching behaviors induced by intradermal

chloroquine injection. The plots indicate the cumulative number of scratching bouts recorded every 5 min. **e** Optical inhibition of ORX neurons reduced the chloroquine-induced scratching behaviors (*n* = 7–8 mice in each group). **f** Time course of the number of wiping behaviors induced by capsaicin injection into the cheek. The plots indicate the cumulative number of wiping events recorded every 3 min. **g** Optical inhibition of ORX neurons increased the capsaicin-induced wiping behaviors (*n* = 8–9 mice in each group). The data represent the mean ± SEM. **p < 0.01, ***p < 0.001; two-way ANOVA with Tukey's multiple comparisons tests. ORX, orexin; tTA, tetracycline transactivator; TRE, tetracycline response element; LH, lateral hypothalamus; Aq, aqueductal; lPAG, lateral periaqueductal gray; vlPAG, ventrolateral periaqueductal gray.

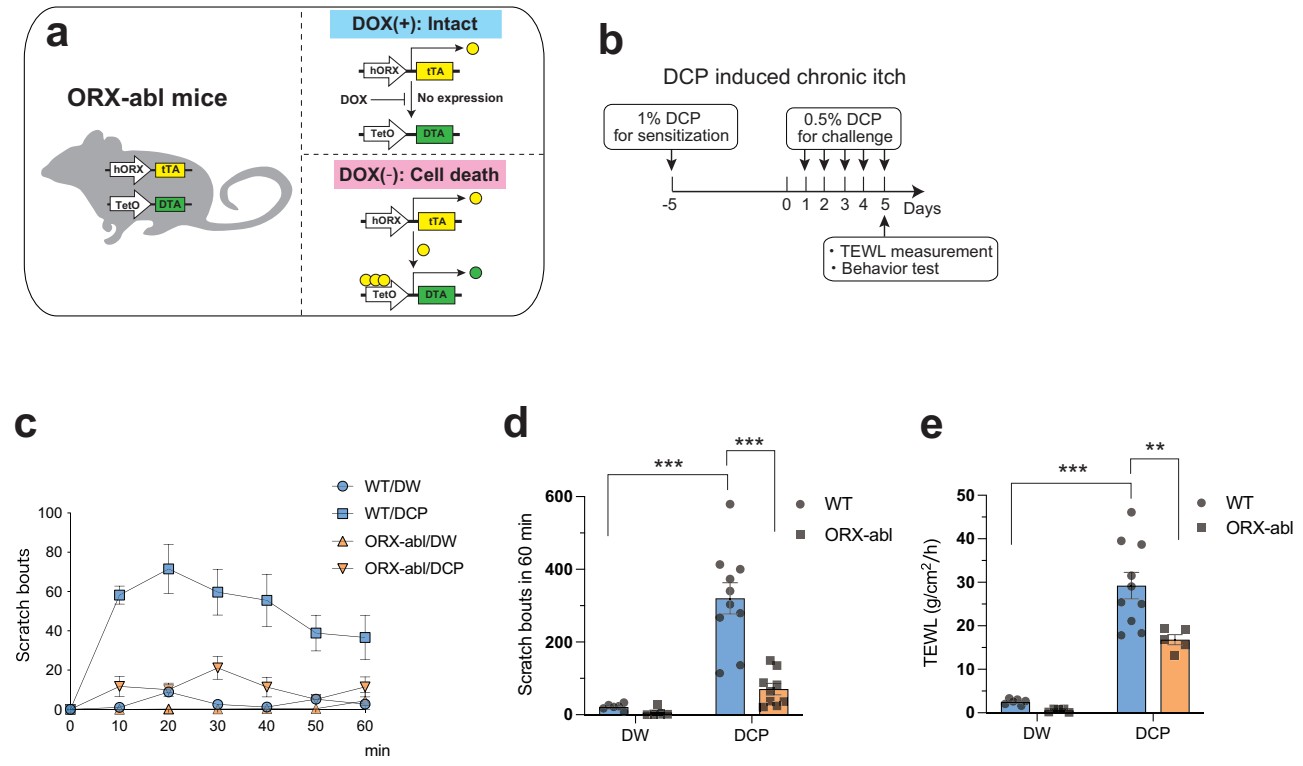

**Fig. 4 | Ablation of orexin neurons suppresses scratching behavior and skin barrier dysfunction in chronic pruritus model. a** Schematic illustration showing the specific ablation of orexin neurons using tet-off system in ORX-abl mice. In the absence of DOX, orexin neurons are specifically ablated by expressing DTA. **b** Experimental timeline of the DCP-induced chronic pruritus model. **c** Time course of the number of scratching behaviors in DCP-induced chronic itch model. The plots indicate the cumulative number of scratching bouts recorded every 10 min. **d** Scratching behaviors during DCP-induced chronic itch were significantly

suppressed in ORX-abl mice (*n* = 5–10 for each group). **e** The rise of TEWL by repeated DCP application was decreased in ORX-abl mice compared to WT counterparts (*n* = 5–10 for each group). The data represent the mean ± SEM. **p < 0.01, ***p < 0.001; two-way ANOVA with Tukey's multiple comparisons test. hORX promoter, human prepro-orexin promoter; tTA, tetracycline transactivator; DTA, diphtheria toxin A; DOX, doxycycline; DCP, diphenylcyclopropenone; TEWL, transepidermal water loss; DW, distilled water.

inhibition of Tac1-positive glutamatergic neurons in the PAG results in reduced pruritogen-induced scratching behaviors, indicating the vital role of these neurons in facilitating itch neural processing. In contrast, the ablation of SST-positive glutamatergic neurons in the PAG was found to have no impact on itch neural processing. In our previous studies[2,11], utilizing the *cFos-tTA; TetO-GCaMP6* mice, we demonstrated that a majority of ORX neurons were activated by both nociceptive and pruritic stimulation. This finding corroborates the notion that a substantial proportion of ORX neurons are dually responsive to nociceptive and pruritic stimuli, potentially exerting dichotomous modulatory effects on these two sensations. Additionally, in experiments with orexin-peptide-deficient mice (ORX-KO), no significant deviation in scratching behavior was observed in comparison with wild-type (WT) controls[2,11], in contrast to the results using ORX-abl mice and optogenetics. These results suggest that the synaptic transmission by the orexin peptide itself (orexin peptide-dependent synaptic transmission) is not

necessary for facilitating itch processing. Instead, co-transmitters expressed in ORX neurons, such as glutamate[36], dynorphin[37], or neurotensin[38], might have a significant role in instigating the itch neural pathway. Contrarily, orexin-dependent transmission has been reported to activate the analgesic effect by ORX neurons[39,40]. Previous research reported orexin-dependent and -independent functions of ORX neurons in regulating sleep[41] and body temperature[42] respectively, suggesting the potential for the transmitter-dependent functional segregation within the same ORX neuron in pain and itch neural processing. Thus, integrating these prior discoveries with our current findings, it could be hypothesized that orexin-independent synaptic transmission by ORX neurons might activate Tac1-positive PAG glutamatergic neurons and instigate itch neural processing. Conversely, orexin-dependent synaptic transmission to other subpopulations of PAG glutamatergic neurons, such as SST-positive neurons, may play a role in pain relief processing via ORX neurons. Nevertheless, prior to drawing definitive

**Article**

**Fig. 5 | The LH → PAG axis of orexin neurons is crucial for the diametric regulation of itch and pain processing.** A visual summary of our present findings. The ArchT-driven optical inhibition of LH$^{ORX}$ → lPAG/vlPAG neuronal terminals resulted in a decrease in itch-related behavior and an increase in pain-related behavior (Fig. 3). This result aligns with those obtained from optogenetic inhibition at the cell body of ORX neurons in the LH (Fig. 1). Thus, our present findings support a pivotal role for the LH → PAG axis projection of ORX neurons in the opposing modulation of itch and pain neural processing. LH, lateral hypothalamus; lPAG, lateral periaqueductal gray; vlPAG, ventrolateral periaqueductal gray.

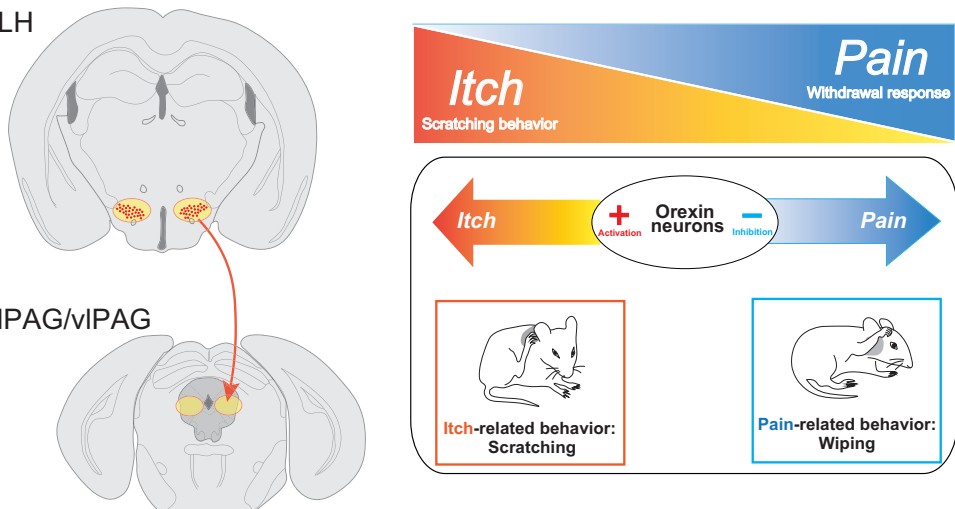

conclusions, several aspects warrant additional investigation. It is postulated that Tac1+ PAG glutamatergic neurons may be heterogeneous, comprised of multiple subpopulations endowed with distinct functionalities beyond mediating scratching[35]. Future research endeavors are required to classify l/vlPAG Tac1+ neurons by their molecular signatures and/or their synaptic connectome to elucidate the array of their functional roles, as well as to delineate their interactions with ORX neurons. Following the classification of Tac1+ neuron subgroups, it will be imperative to map the expression patterns of ORX receptors across the glutamatergic neurons within the PAG, particularly within the various Tac1-positive neuron subsets, to fully comprehend their neurobiological significance. Therefore, further investigation is still required to delineate the precise neural circuit mechanism responsible for the contrasting regulation of pain and itch by ORX neurons.

Acute pruritus serves as a protective mechanism, alerting the body to external threats and triggering a scratching response to eliminate potential irritants; this response can be perceived as a defensive action. Our research has thus far established that ORX neurons are instrumental in the neural processing of acute itch, as demonstrated through the chloroquine-induced acute pruritus model (Figs. 1–3). However, when the acute itch condition transitions to a chronic state, its protective function deteriorates, transforming into a persistent and challenging disease that considerably reduces a patient's quality of life due to a relentless itch-scratch vicious cycle[43–45]. Chronic pruritus has been linked to a range of medical conditions, such as inflammatory skin diseases[46,47], chronic kidney disease[48], intrahepatic cholestasis[49,50], diabetic neuropathy[51,52], post-herpetic itch[53], and psychogenic itch[54]. Despite its high prevalence and considerable socioeconomic burden, optimal treatment for chronic itch remains elusive[55,56]. One of the reasons for this unmet medical need is the need for a comprehensive understanding of the precise neural mechanisms implicated in different types of chronic pruritus. Over the past few decades, key molecules, neurons, and neural pathways specific to acute itch have been progressively uncovered[14,57,58]. However, compared to acute itch, the pathological mechanisms of chronic itch have been relatively understudied, potentially hindering progress in clinical treatment. In this study, we provided evidence that ORX neurons also play a role in modulating scratching behaviors in a chronic contact dermatitis mouse model (Fig. 4). The scratching behaviors induced by the repetitive application of DCP to the nape skin were significantly suppressed in ORX-abl mice compared to the WT control group. Furthermore, ORX-abl mice exhibited significantly lower TEWL values, a reliable measure for evaluating skin barrier function, than the WT control group. Our findings indicate that suppressing ORX neurons may reduce scratching behavior and subsequently prevent skin barrier dysfunction caused by chronic itch. As such, ORX neurons are implicated in itch neural processing in both acute and chronic itch situations. This highlights the potential therapeutic relevance of targeting ORX neurons in managing chronic itch conditions such as atopic dermatitis.

Accumulating empirical data also substantiate the involvement of ORX neurons in several chronic pain models, encompassing chronic neuropathic pain, diabetic neuropathic pain, and rheumatoid arthritis (RA)-induced hyperalgesia[59]. Chronic neuropathic pain, delineated by persistent pain, hyperalgesia, and allodynia, has been shown to be mitigated by orexin-A through intrathecal or intracerebroventricular administration, as demonstrated in models of partial sciatic nerve ligation[60]. Another study revealed that orexin attenuated thermal hyperalgesia following chronic constriction injury (CCI) of the sciatic nerve in rat[61]. In the context of diabetic neuropathic pain, intrathecal administration of orexins yielded an antinociceptive effect on diabetic neuropathic pain in streptozotocin-induced diabetic rats[62]. Furthermore, RA, characterized by joint damage, hyperalgesia, and cachexia, has been responsive to the administration of orexin-A, evidenced by alleviated hyperalgesic symptoms and improved cachexia in an adjuvant-induced arthritis (AIA) rat model with daily intravenous orexin-A injections over an 8-day course[63]. Consequently, our present findings regarding chronic pruritus, alongside extant literature on chronic pain, illuminate the potential therapeutic implications of ORX neuron targeting in the management of chronic itch and pain.

In conclusion, our research shows the opposing roles of ORX neurons in itch and pain neural processing. Our present findings offer a fresh lens to understanding the central neural mechanism and the physiological meaning of the antagonistic regulation of itch and pain sensation. This is just the preliminary step towards discovering and advancing innovative therapies for managing pain and itch, so we remain dedicated to pursuing additional research to advance our understanding in this field further.

## Methods
### Ethics
All experiments followed the guidelines outlined by the Physiological Society of Japan and were approved by the Experimental Animal Research Committee of Kagoshima University (MD19087, MD19100, MD21083, MD22042, MD22056, MD22109).

### Animals
Experiments were performed using wild-type mice (C57BL/6, male, Japan CREA), ORX-tTA mice (male), and ORX-abl mice (male). All animals were 8–16 weeks old at the start of each experiment.

Orexin-tetracycline transactivator (tTA) mice (ORX-tTA) express tTA exclusively in orexin neurons under the control of the human prepro-orexin promoter[64].

Regarding the method for selective ablation of orexin neurons (ORX-abl), ORX-tTA mice were bred with tetO diphtheria toxin A fragment (DTA) mice (B6.Cg-Tg (tetO DTA) 1Gfi/J, The Jackson Laboratory) to generate orexin-tTA; tetO DTA mice. In these double-transgenic mice, doxycycline is removed from their chow starting from birth, so by 4 weeks of age, almost all (> 97%) of orexin neurons are ablated[64].

Mutant mice were maintained as heterozygotes and crossed to obtain null mutants. We backcrossed the mutant mice with C57BL/6 mice (Clea Japan Inc., Tokyo, Japan) for over 10 generations. We have confirmed that the expression of orexin peptide disappeared in ORX-abl mice[28,42] and also confirmed in the present experiment (Supplementary Fig. 2).

Animals were housed with lights on at 7:00 am and off at 7:00 pm. All experiments were performed during the light cycle, i.e., between 10:00 am and 6:00 pm. All experiments followed the guidelines outlined by the Physiological Society of Japan and were approved by the Experimental Animal Research Committee of Kagoshima University.

## Viral constructs and stereotaxic surgery
The following viruses were used: AAV-TRE-ArchT-mCherry (Serotype DJ/8, titer $1.0 \times 10^{13}$ genomic copies per ml), AAV-TRE-ChR2-mCherry (Serotype DJ/8, titer $3.3 \times 10^{13}$ genomic copies per ml), AAV-TRE-mCherry (Serotype DJ/8, titer $1.1 \times 10^{12}$ genomic copies per ml) were generated and purified in the Okuno lab at Kagoshima University.

Briefly, cDNAs for eArchT3.0 and hChR2(H134R) were obtained as a gift from Dr. Karl Deisseroth via Addgene (#35513 and #20297) and sub-cloned into an AAV ITR plasmid with the TRE promoter and mCherry cDNA. The resultant AAV plasmids were cotransfected with pAAV-DJ/8 and pHelper Vectors (Cell Biolabs, VPK-400-DJ-8) into HEK293 cells. After extracted from the cells, AAV particles were purified by iodixanol discontinuous density gradient, and were buffer-exchanged and concentrated using Amicon Ultra ultrafiltration (Merck). The titers of viruses were determined by qPCR.

Stereotaxic surgeries were performed as previously described[30]: The mice were anesthetized with the inhalation of 3% isoflurane using a stereotaxic instrument (ST-7, Narishige, Tokyo, Japan) and given an analgesic (buprenorphine, 0.05 mg/kg) and an antibiotic (penicillin G, 40,000 U/kg) agent subcutaneously. Body temperature was kept stable by using a heating pad during surgery. The skull was exposed with a small incision, and holes were drilled to inject the virus with glass micropipettes (50 μm in diameter at the tip). Viral injections were targeted into the LH using coordinates (AP 1.4 mm, ML ± 1.0 mm, DV 5.5 mm from bregma) based on the Paxinos and Franklin mouse brain atlas (2nd edition). For each side, 300 nL of the virus was injected over 10 min with a gas-pressure microinjector (BJ-110, BEX, Japan) attached to a micropipette with a silicone tube. After the injection, the pipette was left in place for about 5 min before retraction. After surgery, the animals were placed on a 37 °C warm plate for recovery.

Three weeks after viral injection, optical fibers (0.2 mm in core diameter, 0.22 NA; KYOCERA, Kyoto, Japan) were placed 1 mm above the viral injection site in the LH for optogenetic experiments. For the optical terminal inhibition experiment (Fig. 3), optical fibers were bilaterally implanted into lPAG/vlPAG region using coordinates (AP 4.4 mm, ML ± 1.0 mm, DV 2.8 mm from bregma, angle ± 8°). Optical fibers were secured with dental cement and skull screws. Viral injection and fiber implantation sites were confirmed post hoc in all animals (Supplementary Fig. 1), and those with incorrect locations were excluded from the final analyses.

## Behavioral experiments
In all behavioral experiments, mice were handled for at least 5 days before performing the behavioral experiments. All animals were acclimatized to the observation chamber for 3 days before the behavioral experiments.

## Acute itch model (neck model)
The pruritogen-induced neck-scratching model was performed as described previously[11,18]. The nape of the neck of the mice was shaved the day before the experiment. The animals were placed individually in the observation chamber and allowed to habituate to it for 30 min. Chloroquine (200 μg/50 μL) was injected intradermally into the nape, and the mice were immediately placed into the observation chamber. Subsequently, the scratching behaviors were video recorded at 60 frames/s for 30 min in an unmanned environment, and the video was then played back to assess the scratching behavior. The scratching behavior was quantified by counting the number of scratching bouts, consisting of one or more rapid back-and-forth hind-paw motions on the intradermal injection site. Counting of scratching behavior was performed in a blinded manner.

## Acute pain model (cheek model)
The cheek-pain model was performed as described elsewhere[11,65,66]. The right cheek of the mice was shaved the day before the experiment. Animals were moved to the recording cage for 30 min to acclimatize to the recording conditions. The mice were then gently handheld, and capsaicin (40 μg/20 μL) was injected intradermally into the right cheek. Subsequently, the wiping behaviors (pain-related behavior) were video-recorded for 30 min in an unmanned environment. The video was then played back to count the number of wiping of the injecting site. The analytical timeframe was set to an initial 15 min after injection. To distinguish pain-related wiping (typically wiping the injection site using the ipsilateral forepaw) from grooming (typically wiping using both forepaws), we counted the number of wiping events using a single ipsilateral forepaw. All analyses of behavioral assessment were performed in a blinded manner.

## Optogenetic manipulations during acute behavioral tests
The optical fiber was connected to a 473 nm or 532 nm laser power source (473 nm laser: BL473T8-100FC, Shanghai Laser & Optics Century Co., Ltd., China, 532 nm laser: GL532T3-300FC, Shanghai Laser & Optics Century Co., Ltd., China) to the implanted optical fiber in the mice with attaching to a rotary joint (FRJ_1 × 1_FC-FC, Doric Lenses, Quebec, Canada) to allow the mice for freely moving. The laser was controlled with a stimulator (SEN-3301, NIHON KOHDEN, Tokyo, Japan).

For optogenetic inhibition of ORX neurons, green light (532 nm) was delivered continuously at 3 min intervals (3 min light-on period following a 3 min light-off period), repeated 5 times over the total period of 30 min after the injection of pruritogen or algogen. The final output power for the light ranged between 7-9 mW depending on the light transmission efficacy of the optical fiber used.

For optogenetic stimulation of ORX neurons, blue light (473 nm) was delivered in 10 ms pulses at 20 Hz, with the 3 min light-on period following a 3 min light-off period, repeated 5 times over the total period of 30 min after the injection of pruritogen or algogen. The final output power for the light ranged between 6-8 mW depending on the light transmission efficacy of the optical fiber used.

To examine the c-Fos expression of ORX neurons by ChR2-induced optical stimulation, 1.5 hours after the optical stimulation over 30 min, the mice were deeply anesthetized with urethane (1.3 g/kg, i.p.) and transcardially perfused with saline followed by 4% paraformaldehyde in 0.01 M PBS (pH 7.4). Then brains were excised and processed under immunohistochemical experiment as described in the section of immunohistochemistry, below.

## Chronic itch model (diphenylcyclopropenone (DCP)-induced model)
A chronic contact dermatitis-related chronic itch model was made by applying DCP onto the neck skin as previously described[15,18,20,21]. The nape of the neck of the mice was shaved, and the shaved area was topically applied 0.2 mL of 1% DCP (dissolved in acetone) for initial sensitization. Five days after the initial sensitization, the applicated area was challenged by painting with 0.2 mL of 0.5% DCP for 5 days. Distilled water was applied as a control agent. Scratching behaviors were video recorded for 60 minutes after the final DCP challenge. The scratching behaviors were analyzed as described in the acute itch model.

**Article**

Transepidermal water loss (TEWL) was measured before the final challenge of DCP on day 5, under isoflurane anesthesia using a Tewameter TM210 (Courage & Khazawa, Cologne, Germany), as described[67] previously.

## Immunohistochemistry

Animals were deeply anesthetized with urethane (1.3 g/kg, i.p.) and transcardially perfused with saline, followed by 4% paraformaldehyde in 0.01 M PBS (pH 7.4). The brain was excised, post-fixed at 4 °C overnight, and cryoprotected with 30% sucrose in 0.01 M PBS. Subsequently, 40-μm sections were prepared using a cryostat (Microtome Cryostar NX70, Thermo Fisher Scientific). Every fourth section was collected, and immunohistochemical staining of floating sections was performed. The sections were incubated with PBS containing 0.3% Triton-X and 1% normal horse serum for 30 min.

For orexin staining, sections were reacted with a rabbit anti-orexin A serum (14346-v, 1/500, PEPTIDE INSTITUTE, INC.) for 90 min at room temperature (RT). After washing with PBS three times, the sections were incubated with secondary antibodies (CF488-conjugated anti-rabbit IgG, 1/500, Biotium) for 60 min in a dark box at RT. After rewashing three times, the sections were mounted on a glass slide and examined under a fluorescence microscope (BZ-X700, KEYENCE, Osaka, Japan). We counted the number of mCherry-expressing cells and orexin-immunoreactive cells identified as ORX neurons. The numbers of mCherry/ORX co-labeling neurons were counted from 3 slices per mouse centered on the LH coordinates.

For confirming the c-Fos expression of ORX neurons by ChR2-induced optical stimulation, the sections were stained with a rabbit anti-c-Fos monoclonal antibody (9F6, 1/250, Cell Signaling Technology) for 90 min at RT. After washing with PBS, the sections were incubated with secondary antibodies (CF750-conjugated anti-rabbit IgG, 1/500, Biotium) for 60 min in a dark box at RT. The sections were then rinsed with PBS and reacted with a guinea pig anti-orexin A serum (389004, 1/250, Synaptic systems) for 60 min at RT. After rinsing with PBS, the sections were incubated with secondary antibodies (CF488-conjugated anti-guinea pig IgG, 1/500, Biotium) for 60 min in a dark box at RT. We counted the numbers of c-Fos/ORX co-labeling neurons were counted from 3 slices per mouse centered on the LH coordinates.

For confirming the mCherry-expressing axons in the PAG area, the sections were stained with a goat anti-mCherry polyclonal antibody (AB0081-200, 1/500, SICGEN ANTIBODIES) for 60 min at RT. After rinsing with PBS, the sections were incubated with secondary antibodies (CF568-conjugated anti-goat IgG, 1/500, Biotium) for 60 min in a dark box at RT.

For the confirmation of orexin neuron ablation in ORX-abl mice in a chronic itch experiment (Fig. 4 and Supplementary Fig. 3), the sections were stained with a guinea pig anti-orexin A serum (389004, 1/250, Synaptic systems) for 60 min at RT. After rinsing with PBS, the sections were incubated with secondary antibodies (CF568-conjugated anti-guinea pig IgG, 1/500, Biotium) for 60 min in a dark box at RT. The numbers of ORX-positive cells were counted from every 4th section in an animal (six sections per mouse), as orexin neurons were distributed over a rostrocaudal distance of ~1 mm centered on the LH coordinates[23,42].

## Statistical analyses

No statistical methods were used to pre-determine sample sizes, but our sample sizes are similar to those in previous reports and are typical of the field[6,18,35].

Statistical analyses were performed via a paired and unpaired t-test or two-way ANOVA with post hoc Tukey's multiple comparisons test using the Prism9 software (GraphPad Software, San Diego, CA, USA). All data are presented as mean ± standard error of the mean (SEM), with n indicating the number of mice analyzed. Statistical significance was set at $P < 0.05$ in all analyses.

## Reporting summary

Further information on research design is available in the Nature Portfolio Reporting Summary linked to this article.

## Data availability

The source data behind the graphs in the paper can be found in Supplementary Data 1. Additional data supporting the findings of this study are available from the corresponding author upon request.

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

## Acknowledgements

This work was supported by JSPS KAKENHI (26670290, 23K06015, 20H05068, 23H02590, 22K07330, 16H05130) and Kodama Memorial Fund for Medical Research. The authors thank the Departments of Physiology members for the fruitful discussion and technical advice. We also acknowledge the Joint Research Laboratory, Neuroscience Core, and Laboratory of Animal Science at the Kagoshima University Graduate School of Medical and Dental Sciences for using their facilities.

## Author contributions

Tatsuroh Kaneko: Designed experiments, performed experiments, analyzed data, and wrote the manuscript. Asuka Oura: Performed experiments, analyzed data, and manuscript feedback. Yoshiki Imai: Performed experiments, analyzed data, and manuscript feedback. Ikue Kusumoto-Yoshida: Technically advised and supported for optogenetic experiments, and manuscript feedback. Takuro Kanekura: Technically advised and supported for chronic itch experiments, and manuscript feedback. Hiroyuki Okuno: Technically advised and supported for optogenetic experiments, and manuscript feedback. Tomoyuki Kuwaki: Designed experiments, analyzed data, and wrote the manuscript. Hideki Kashiwadani: Designed experiments, analyzed data, and wrote the manuscript. All authors approved the final version of the manuscript.

## Competing interests

The authors declare no competing interests.
