## [Peer Review File · Communications Biology]

Reviewers' comments:

Reviewer #1 (Remarks to the Author):

This study revealed that the role of the orexin-producing neurons in the LH, which regulated pain and itch processing bidirectionally in chloroquine- and capsaicin-induced mice models. In addition, the pathway from LHORX to IPAG/vIPAG participated the regulation. Results also provided new insights to elucidate the potential mechanism of chronic itch. The article is interesting and, but there are some issues regarding results and discussion.

Major concern:

1. In the result 2, optogenetic activation of ORX neurons did not affect acute itch and pain behaviors.

Author also mentioned in the discussion (lines 244-254) that was due to external stimulant that increased the firing rates of the LHORX neurons to the maximum. Optogenetic activation could not further elevate the level of neuronal activity. This hypothesis could be verified by an experiment by comparing the expression of c-Fos (or other markers of neuronal activity) in LHORX neurons under light-on and light-off conditions. If there is no difference, it indicates that optogenetic activation cannot further improve its activity.

2. In the result 3, inhibition of the LHORX to the IPAG/vIPAG pathway suppressed the itch-related, but enhanced the pain-related behaviors. This phenomenon was explained to be mediated by two subsets of neurons (Tac+ and SST+) in the PAG, respectively in the discussion (lines 285-291). So that, whether these two types of neurons have clear anatomical features with LH, and whether there is differential expression of ORX receptors, should be further discussed.

3. In the result 4, authors utilized the ORX-ablation mice to demonstrate the important role of the LHORX neurons in DCP-induced chronic pruritus model and provided the potential therapeutic target in clinical settings. However, it is wondering that the LHORX neurons also mediate the chronic pain.

Minor concern:

1. In figure 3G, the level of wipes in Light OFF and mCherry groups seem lower than those in figure 2G and figure 1I.

2. In figure 1H, the difference between Arch-Light ON group and other groups on 5-min seems remarkable than other timepoints. Whether the regulation of the LHORX neurons had the time effect?

Reviewer #2 (Remarks to the Author):

Pain and itch are widely believed to interact closely, however, the precise neural mechanisms are far more from being fully understood. In this manuscript, Kaneko et al., demonstrated that a cluster of ORX neurons in the lateral hypothalamus involved in both pain perception and itch regulation, and their projections to the periaqueductal gray may be as novel neural substrates for pain-itch interactions. To maximize the clinical relevance, the authors also evaluated the regulatory effect of these cells in a chronic itch model that mimics atopic dermatitis in mice. The topic is interesting and important and the manuscript was well written. There are, however, some concerns that need to be addressed before considering it for publication. The detail as following:

1. In this study, the authors employed two itch-related models, encompassing both acute and chronic itch, to fully demonstrate the role of the ORX signal in itch modulation. However, the absence of experiments involving a chronic pain model in this study somewhat weakens the evidences related to pain. Furthermore, it appears that capsaicin-induced acute pain is only sustained for several minutes (Fig.1H, Fig.2F), suggesting that analyzing statistical data over 30 minutes may lead to potential false positive or false negative conclusions. It may be more appropriate to restrict the analysis to data collected from 0 to 15 minutes.
2. Fig.1F: The number of scratch bouts during Light OFF period in the ArchT group appears significantly higher than those of the other CONTROL treatment (mCherry group). Please clarify.
3. Optogenetics and chemogenetics offer complementary advantages. Given the relatively persistent itch process induced by chloroquine and the experimental design, it raises the question of why the authors chose optogenetic techniques instead of longer-acting chemogenetic regulation of ORX neurons.
4. The study reported the dual functions of LH-ORX neurons in the sensory domain. Prior to establishing the association between ORX neurons and pain-itch interactions, the authors should initially furnish data concerning the individual neural response of LH-ORX cells to pain and itch stimulation, which can be conveniently obtained through techniques such as c-Fos immunofluorescence staining or other more immediate methods. Additionally, the reviewer has raised the question of whether the cells responding to both stimuli are the same population of neurons within the LH, particularly, whether the majority of these cells are the same ORX neurons mentioned here.
5. Based on the behavioral data, the distribution of scratch bouts and wipes in mice appears to be relatively dispersed, making a sufficient sample size crucial. Regrettably, in some groups (Fig. 2E, Fig. 2F, Fig. 3G), a sample size of 5 may be somewhat underpowered.
6. While optical inhibition of orexin neurons eliciting a contrasting modulation in itch and pain, optical activation of these neurons unaltered the two sensations. The authors ascribed it to a “ceiling effect” reasonably. Despite the comprehensive discussion in lines 100-122, experimental results are still desirable. For instance, it would be valuable to determine whether the number of c-Fos cells under pruritogen conditions is likely to be equal to that in mice subjected to additional optogenetic activation.
7. In this story, the LH→PAG projection functions as a circuit mechanism through which ORX neurons exert their influence on pain and itch regulation. However, the investigation regarding the neural circuitry was conducted in a rudimentary manner here. If possible, it is recommended to include the

identification of downstream targets within the periaqueductal gray with respect to cell type specificity (glutamatergic or GABAergic neurons) and receptor type specificity (ORX-1R or -2R).

Reviewer #1 (Remarks to the Author):

This study revealed that the role of the orexin-producing neurons in the LH, which regulated pain and itch processing bidirectionally in chloroquine- and capsaicin-induced mice models. In addition, the pathway from LHORX to IPAG/vIPAG participated the regulation. Results also provided new insights to elucidate the potential mechanism of chronic itch. The article is interesting and, but three are some issues regarding results and discussion.

Major concern:

1. In the result 2, optogenetic activation of ORX neurons did not affect acute itch and pain behaviors. Author also mentioned in the discussion (lines 244-254) that was due to external stimulant that increased the firing rates of the LHORX neurons to the maximum. Optogenetic activation could not further elevate the level of neuronal activity. This hypothesis could be verified by an experiment by comparing the expression of c-Fos (or other markers of neuronal activity) in LHORX neurons under light-on and light-off conditions. If there is no difference, it indicates that optogenetic activation cannot further improve its activity.

REPLY:

We extend our gratitude for the insightful suggestions during your busy schedule.

Pursuant to your suggestions, we have executed additional experiments to ascertain the degree of activation of ORX neurons upon optical stimulation, assessed via c-Fos immunohistochemistry.

These additional investigations verified an augmented c-Fos expression in ORX neurons following optical activation, with a statistically significant increase compared to the baseline levels observed in the absence of light stimulation (Line: 119-127, and Supplementary Figure 2).

Notably, the number of c-Fos positive cells elicited by optical stimulation was comparable to that induced by chloroquine-induced pruritic stimuli (Line: 123-127), as reported in our previous study (Kaneko et al. J. Physiol. Sci. 72, (2022)).

Consequently, these findings are in alignment with our postulated hypothesis that sensory activation has already maximized the responsiveness of ORX neurons corresponding to pruritic or nociceptive inputs, rendering them unresponsive to further stimulation via optogenetic activation.

Your constructive suggestion was instrumental in examining our hypothesis with experimental data. We express our sincere appreciation.

2. In the result 3, inhibition of the LHORX to the IPAG/vIPAG pathway suppressed the itch-related, but enhanced the pain-related behaviors. This phenomenon was explained to be mediated by two subsets of neurons (Tac+ and SST+) in the PAG, respectively in the discussion (lines 285-291). So that, whether these two types of neurons have clear anatomical features with LH, and whether there is differential expression of ORX receptors, should be further discussed.

REPLY:

We are grateful for your insightful recommendation, which we acknowledge as pivotal for guiding subsequent inquiries within this domain.

We refer to the seminal work by Gao et al. (*Neuron* 101, 45-59.e9, 2019), who underscored the significance of Tac1+ PAG glutamatergic neurons in itch processing. In their literature, they also proposed that Tac1+ neurons exhibit heterogeneity, potentially constituting discrete subgroups with varied functional roles extending beyond itch processing. Notably, the activation of I/vIPAG Tac1+ neurons was associated with pronounced grooming behaviors distinct from scratching, and influenced motor coordination and anxiety-related behaviors, as evidenced by alterations in rotarod performance and open-field test metrics.

Accordingly, a prerequisite to further investigation is the characterization and classification of these Tac1+ PAG neuron subpopulations. Subsequent to this, we can proceed to analyze the expression patterns of ORX receptors among PAG glutamatergic neurons.

We have integrated this nuanced complexity into our discussion, amending the discourse to better reflect the intricate nature of this research area (Line: 326-334).

The depth and direction of our discussion have been significantly enhanced by your constructive feedback, for which we are immensely thankful.

3. In the result 4, authors utilized the ORX-ablation mice to demonstrate the important role of the LHORX neurons in DCP-induced chronic pruritus model and provided the potential therapeutic target in clinical settings. However, it is wondering that the LHORX neurons also mediate the chronic pain.

REPLY:

Thank you very much for your insightful comments.

The role of ORX neurons in mediating chronic pain has been previously explored by other investigators. However, the specific involvement of ORX neurons in itch processing has not been investigated, and our findings represent a novel insight into the role of ORX neurons in the neural mechanisms of itch. Consequently, we have investigated the influence of ORX neurons on chronic pruritus employing DCP-induced chronic pruritus model, alongside the chloroquine-induced acute model, to elucidate their potential clinical relevance.

Nonetheless, as per your comment, we didn't describe the involvement of ORX neurons in chronic pain processing. Accordingly, we have meticulously detailed the engagement of ORX neurons in various chronic pain conditions, including chronic neuropathic pain, diabetic neuropathic pain, and rheumatoid arthritis-associated pain, integrating appropriate references (Line: 369-384).

Your constructive feedback has been instrumental in enhancing the depth of our discussion on this topic. We are truly thankful for your valuable suggestions.

Minor concern:

1. In figure 3G, the level of wipes in Light OFF and mCherry groups seem lower than those in figure 2G and figure 1I.

REPLY:

Thank you for your comment.

Initially, the mCherry-control groups were constituted by a small sample size (n=5), which yielded a relatively high variance in the number of wipes. In response to this concern, we have undertaken additional experiments to augment the sample size of these control groups to 8-9 subjects (n= 5 → 8-9). Subsequent to these additional experiments, the dispersion of wipes has been observed to align more closely with the data presented in Figures 1I and 2G.

2. In figure 1H, the difference between Arch-Light ON group and other groups on 5-min seems remarkable than other timepoints. Whether the regulation of the LHORX neurons had the time effect?

REPLY:

We are grateful for your insightful feedback. Following your observations, the difference in the number of wipes between the Arch-Light ON group and other

groups was remarkable in the first half of the observation period of 30 minutes. Given the transient nature of capsaicin-induced acute nociception, we postulated that a prolonged observational period (30 minutes) might obfuscate the true temporal dynamics of acute pain responses. Consequently, we refined our methodological approach by reducing the analytical timeframe to 15 minutes, thereby improving the temporal fidelity of our data (Figure 1H, 2F, and 3F). The refinement of the analytical timeframe revealed a sustained attenuation of nociceptive behavior in ORX neuron-suppressed subjects through optogenetics compared to the control groups.

Your suggestions were pivotal in augmenting both the granularity of our analysis and our comprehension of the underlying phenomena. We really appreciate your constructive comments for the refinement of our study.

Reviewer #2 (Remarks to the Author):

Pain and itch are widely believed to interact closely, however, the precise neural mechanisms are far more from being fully understood. In this manuscript, Kaneko et al., demonstrated that a cluster of ORX neurons in the lateral hypothalamus involved in both pain perception and itch regulation, and their projections to the periaqueductal gray may be as novel neural substrates for pain-itch interactions. To maximize the clinical relevance, the authors also evaluated the regulatory effect of these cells in a chronic itch model that mimics atopic dermatitis in mice. The topic is interesting and important and the manuscript was well written. There are, however, some concerns that need to be addressed before considering it for publication. The detail as following:

1. In this study, the authors employed two itch-related models, encompassing both acute and chronic itch, to fully demonstrate the role of the ORX signal in itch modulation. However, the absence of experiments involving a chronic pain model in this study somewhat weakens the evidences related to pain. Furthermore, it appears that capsaicin-induced acute pain is only sustained for several minutes (Fig.1H, Fig.2F), suggesting that analyzing statistical data over 30 minutes may lead to potential false positive or false negative conclusions. It may be more appropriate to restrict the analysis to data collected from 0 to 15 minutes.

REPLY:

Thank you very much for your insightful comments during your busy schedule. Below are our correspondences to your suggestions.

[Chronic pain model]

The role of ORX neurons in mediating chronic pain has been previously explored by other investigators. However, the specific involvement of ORX neurons in itch processing has not been investigated, and our findings represent a novel insight into the role of ORX neurons in the neural mechanisms of itch. Consequently, we have investigated the influence of ORX neurons on chronic pruritus employing DCP-induced chronic pruritus model, alongside the chloroquine-induced acute model, to elucidate their potential clinical relevance.

Nonetheless, as per your comment, we didn't describe the involvement of ORX neurons in chronic pain processing. Accordingly, we have meticulously detailed

the engagement of ORX neurons in various chronic pain conditions, including chronic neuropathic pain, diabetic neuropathic pain, and rheumatoid arthritis-associated pain, integrating appropriate references (Line: 369-384).

[Timeframe of capsaicin model]

Following your observations, the difference in the number of wipes between the Arch-Light ON group and other groups was remarkable in the first half of the observation period of 30 minutes. Given the transient nature of capsaicin-induced acute nociception, we postulated that a prolonged observational period (30 minutes) might obfuscate the true temporal dynamics of acute pain responses. Consequently, we refined our methodological approach by reducing the analytical timeframe to 15 minutes, thereby improving the temporal fidelity of our data (Figure 1H, 2F, and 3F). The refinement of the analytical timeframe revealed a sustained attenuation of nociceptive behavior in ORX neuron-suppressed subjects through optogenetics compared to the control groups.

Your suggestions were pivotal in augmenting both the granularity of our analysis and our comprehension of the underlying phenomena. We really appreciate your constructive comments for the refinement of our study.

2. Fig.1F: The number of scratch bouts during Light OFF period in the ArchT group appears significantly higher than those of the other CONTROL treatment (mCherry group). Please clarify.

REPLY:

Thank you very much for the comment.

There were no statistical differences between the ArchT-Light OFF group and other mCherry-control groups in the Two-way ANOVA with Tukey's multiple comparisons test. ($p_{\text{ArchT/Light-OFF vs. mCherry/Light-OFF}} = 0.1606$, $p_{\text{ArchT/Light-OFF vs. mCherry/Light-ON}} = 0.2996$).

We added the results of the statistical test among all groups in the Source data file.

3. Optogenetics and chemogenetics offer complementary advantages. Given the relatively persistent itch process induced by chloroquine and the experimental design, it raises the question of why the authors chose optogenetic techniques instead of longer-acting chemogenetic regulation of ORX neurons.

REPLY:

Thank you very much for your question. We have mainly two reasons why we chose optogenetics rather than chemogenetics: Firstly, our study utilized acute models to assess the involvement of ORX neurons in modulating itch and pain responses. Optogenetics, with its superior temporal resolution, enables rapid and precise modulation of neuronal activity, making it particularly well-suited for acute experimental paradigms where swift activation or inhibition of neuronal populations is required (the rapid temporal modulation facilitates immediate assessment of neuronal functions within specific time frames).

Secondarily, the method of optogenetic terminal inhibition provides a straightforward means to examine specific neural circuits, as elaborated upon in our manuscript (Line: 285-291). Given that ORX neurons form extensive projections to diverse brain regions and thus exhibit multifaceted roles, discerning the specific neural pathways pertinent to itch and pain processing is of paramount interest. The optogenetic methodology permits a facile examination of these circuits by merely altering the location of optical fiber implantation. In our present study, we have expressed the virus vector in ORX neurons of the lateral hypothalamus (LH) while positioning the optical fiber in the periaqueductal gray (PAG), enabling us to validate the significance of the ORX neuronal projections from LH to the PAG.

The inherent simplicity and adaptability of optogenetic techniques, particularly the ability to adapt the implantation site of the optical fiber to probe distinct neural pathways, was a decisive factor in our experiments.

4. The study reported the dual functions of LH-ORX neurons in the sensory domain. Prior to establishing the association between ORX neurons and pain-itch interactions, the authors should initially furnish data concerning the individual neural response of LH-ORX cells to pain and itch stimulation, which can be conveniently obtained through techniques such as c-Fos immunofluorescence staining or other more immediate methods. Additionally, the reviewer has raised the question of whether the cells responding to both stimuli are the same population of neurons within the LH, particularly, whether the majority of these cells are the same ORX neurons mentioned here.

REPLY:

We are grateful for your insightful comments.

In accordance with your suggestion, we have explored the dual responsiveness of ORX neurons to itch and pain stimuli, a critical aspect of our investigation. Previously, we have reported that most ORX neurons react to both nociceptive and pruritic stimuli (Kaneko et al. J. Physiol. Sci. 72, (2022), summarized as follows:

Utilizing *cFos-tTA; TetO-GCaMP6* mutant mice, we have employed a model wherein the transcriptional activator tTA is under the control of the *c-Fos* promoter, becoming active upon neuronal stimulation. This activation facilitates the subsequent binding of tTA to the TetO elements, promoting the expression of the fluorescent protein GCaMP6. The Tet-OFF system employed ensures that in the presence of doxycycline, tTA binding to TetO elements is inhibited, thereby preventing GCaMP6 expression. Conversely, in the absence of doxycycline, GCaMP6 expression is permitted, allowing for the identification of active ORX neurons.

In our experimental paradigm, we induced acute pain in these mice under doxycycline-negative conditions to tag nociceptive-responsive neurons with GCaMP6. Subsequently, we inhibited any new GCaMP6 expression by switching to doxycycline-positive conditions before inducing an itch response to label pruritic-responsive neurons via intrinsic *c-Fos* activation.

The derived images from our previous study indicate a significant augmentation in the proportion of GCaMP6 and c-Fos co-expressing ORX neurons following stimulation compared to controls. A quantitative analysis further corroborated this significant upsurge in GCaMP6 and c-Fos double-positive ORX neurons post-stimulation. A Venn diagram, illustrated in our findings, depicts the relative response of ORX neurons to nociceptive and pruritic stimuli, revealing that a substantial fraction of the ORX neurons are responsive to both pain and itch input.

These findings suggest a considerable overlap in the ORX neuronal populations mediating itch and pain, indicating that the same ORX neurons involved in the processing of both pain and itch sensations.

We have incorporated this discussion into our manuscript, providing a more comprehensive understanding of ORX neurons' functions in itch and pain processing (Line: 305-309).

Your comment was instrumental in enriching the depth of our discussion. We really appreciate it.

5. Based on the behavioral data, the distribution of scratch bouts and wipes in mice appears to be relatively dispersed, making a sufficient sample size crucial. Regrettably, in some groups (Fig. 2E, Fig. 2F, Fig. 3G), a sample size

of 5 may be somewhat underpowered.

REPLY:

Thank you very much for pointing out the issue of inadequate sample sizes within certain experimental groups.

In response to your comments, we have undertaken additional experiments to bolster the sample size in the specified groups, thus enhancing the statistical robustness of our findings. This rectification has resulted in an increase in the number of subjects from an initial $n=5$ to a range of 7-9 individuals, as documented in Figures 2E, 2G, 3E, and 3G.

6. While optical inhibition of orexin neurons eliciting a contrasting modulation in itch and pain, optical activation of these neurons unaltered the two sensations. The authors ascribed it to a “ceiling effect” reasonably. Despite the comprehensive discussion in lines 100-122, experimental results are still desirable. For instance, it would be valuable to determine whether the number of c-Fos cells under pruritogen conditions is likely to be equal to that in mice subjected to additional optogenetic activation.

REPLY:

In accordance with your suggestions, we have executed additional experiments to ascertain the degree of activation of ORX neurons upon optical stimulation, assessed via c-Fos immunohistochemistry.

These additional investigations verified an augmented c-Fos expression in ORX neurons following optical activation, with a statistically significant increase compared to the baseline levels observed in the absence of light stimulation (Line: 119-127, and Supplementary Figure 2).

Notably, the magnitude of c-Fos expression elicited by optical stimulation was comparable to that induced by chloroquine-induced pruritic stimuli (Line: 123-127), as reported in our previous study (Kaneko et al. *J. Physiol. Sci.* 72, (2022)).

Consequently, these findings are in alignment with our postulated hypothesis that sensory activation has already maximized the responsiveness of ORX neurons corresponding to pruritic or nociceptive inputs, rendering them unresponsive to further stimulation via optogenetic activation.

Your constructive suggestion was instrumental in examining our hypothesis with experimental data. We express our sincere appreciation.

7. In this story, the LH→PAG projection functions as a circuit mechanism through which ORX neurons exert their influence on pain and itch regulation. However, the investigation regarding the neural circuitry was conducted in a rudimentary manner here. If possible, it is recommended to include the identification of downstream targets within the periaqueductal gray with respect to cell type specificity (glutamatergic or GABAergic neurons) and receptor type specificity (ORX-1R or -2R).

REPLY:

We express our gratitude for your valuable recommendation, which indeed propels the research focus towards elucidating downstream targets within this domain.

Following your comment 4, it becomes imperative to ascertain whether identical ORX neurons are implicated in both nociceptive and pruritic pathways prior to delving into the downstream neuronal networks and molecular pathways involved. Moreover, discerning the differential employment of neurotransmitters by ORX neurons during itch versus pain processing emerges as an important aspect of this research topic.

In revising the discussion section, we have incorporated an elucidation of our preceding findings indicating that most ORX neurons is responsive to both itch and pain stimuli (Line: 305-309). Furthermore, we have delineated the mechanisms by which orexin peptide-dependent and -independent pathways mediated by ORX neurons facilitate pain and itch processing, respectively, citing potential candidate molecules with references (Line: 309-320).

With regard to PAG, we refer to the seminal work by Gao et al. (*Neuron* 101, 45-59.e9, 2019), who underscored the significance of Tac1+ PAG glutamatergic neurons in itch processing. In their literature, they also proposed that Tac1+ neurons exhibit heterogeneity, potentially constituting discrete subgroups with varied functional roles extending beyond itch processing. Notably, the activation of I/VIPAG Tac1+ neurons was associated with pronounced grooming behaviors distinct from scratching, and influenced motor coordination and anxiety-related behaviors, as evidenced by alterations in rotarod performance and open-field test metrics.

Accordingly, a prerequisite to further investigation is the characterization and classification of these Tac1+ PAG neuron subpopulations. Subsequent to this, we can proceed to analyze the expression patterns of ORX receptors among PAG glutamatergic neurons.

These complexities have been integrated into the discussion section to more accurately mirror the multifaceted nature of this research field (Line: 326-334). Your astute observations have substantially contributed to the refinement and depth of our discourse, for which we express our profound appreciation.

REVIEWERS' COMMENTS:

Reviewer #1 (Remarks to the Author):

The author has already replied to my comments. I have no other questions.

Reviewer #2 (Remarks to the Author):

The authors have answered my concerns and I have no further questions.